# Activation of Purine Biosynthesis Suppresses the Sensitivity of *E. coli gmhA* Mutant to Antibiotics

**DOI:** 10.3390/ijms242216070

**Published:** 2023-11-08

**Authors:** Tatiana A. Seregina, Irina Yu. Petrushanko, Pavel I. Zaripov, Rustem S. Shakulov, Svetlana A. Sklyarova, Vladimir A. Mitkevich, Alexander A. Makarov, Alexander S. Mironov

**Affiliations:** Engelhardt Institute of Molecular Biology, Russian Academy of Science, 119991 Moscow, Russia; waldgang@tuta.io (P.I.Z.); rshakulov@yandex.ru (R.S.S.); sklyarovasveta@yahoo.com (S.A.S.); mitkevich@eimb.ru (V.A.M.); aamakarov@eimb.ru (A.A.M.); alexmir_98@yahoo.com (A.S.M.)

**Keywords:** *∆gmhA* mutants, oxidative stress, supersensitivity, antibiotics, reactive oxygen species, purine biosynthesis, redox protein modification

## Abstract

Inactivation of enzymes responsible for biosynthesis of the cell wall component of ADP-glycero-manno-heptose causes the development of oxidative stress and sensitivity of bacteria to antibiotics of a hydrophobic nature. The metabolic precursor of ADP-heptose is sedoheptulose-7-phosphate (S7P), an intermediate of the non-oxidative branch of the pentose phosphate pathway (PPP), in which ribose-5-phosphate and NADPH are generated. Inactivation of the first stage of ADP-heptose synthesis (*ΔgmhA*) prevents the outflow of S7P from the PPP, and this mutant is characterized by a reduced biosynthesis of NADPH and of the Glu-Cys-Gly tripeptide, glutathione, molecules known to be involved in the resistance to oxidative stress. We found that the derepression of purine biosynthesis (*∆purR*) normalizes the metabolic equilibrium in PPP in *ΔgmhA* mutants, suppressing the negative effects of *gmhA* mutation likely via the over-expression of the glycine–serine pathway that is under the negative control of PurR and might be responsible for the enhanced synthesis of NADPH and glutathione. Consistently, the activity of the *soxRS* system, as well as the level of glutathionylation and oxidation of proteins, indicative of oxidative stress, were reduced in the double *ΔgmhAΔpurR* mutant compared to the *ΔgmhA* mutant.

## 1. Introduction

Violation of the biosynthesis of ADP-glycerol-manno-heptose (ADP-heptose), a component of the cell wall of Gram-negative bacteria, leads to the development of the “deep rough” phenotype, characterized by high sensitivity to hydrophobic compounds including antibiotics [1,2]. In a previous study, we showed that mutant *E. coli* that have lost the ability to synthesize (*∆gmhA*, *∆hldE*, *∆rfaD*) or transfer (*∆waaC* and *∆waaF*) activated ADP-heptose to the inner core of lipopolysaccharides (LPS) develop a powerful oxidative stress, accompanied by the generation of a large number of reactive oxygen species (ROS) and by a diminished pool of reducing equivalents of NADPH [3]. In addition, a change in the normal structure of LPS in bacterial cells leads to radical shifts in the homeostasis of low-molecular thiols (cysteine and hydrogen sulfide) that perform protective functions under the conditions of oxidative stress [3,4,5,6]. Since the enzymes of ADP-heptose biosynthesis are attractive targets for the development of a new generation of antibacterial drugs, a more detailed understanding of the nature of the supersensitivity of “deep rough” mutants to antibiotics is relevant [7,8,9]. The elucidation of processes leading to the suppression of the sensitivity of mutants to antibiotics that have lost ADP-heptose will lead to a better understanding of the nature of metabolic changes taking place in the *gmhA* mutant in order to prevent the emergence of adaptive mechanisms to new drugs.

Biosynthesis of the precursor of activated ADP-heptose, sedoheptulose-7-phosphate (S7P) occurs in one of the central metabolic junction—the pentose phosphate pathway (PPP) [10]. Deletion of the *gmhA* gene encoding sedoheptulose-7-phosphate isomerase prevents the outflow of S7P from the PPP cycle, changing the metabolic equilibrium. The main purpose of PPP is the synthesis of ribose-5-phosphate and the associated generation of NADPH reducing agents [11]. Thus, there is a close relationship between the anabolic processes of synthesis of cell wall components and the synthesis of nucleotide precursors, on the level of which the rate of cell growth and division depends. In this paper, we demonstrate the suppressive effect of activation of purine biosynthesis on antibiotic sensitivity and the redox status of the *E. coli gmhA* mutant.

## 2. Results

### 2.1. Derepression of Purine Synthesis Suppresses the Sensitivity of the gmhA Mutant to Antibiotics Subsection

The processes of synthesis of purine nucleosides and ADP-heptose are related to each other by mutual transformation of PPP intermediates. It has been suggested that an increase in the outflow of ribose-5-phosphate due to the activation of purine synthesis will restore the metabolic equilibrium disturbed as a result of the conservation of S7P during the inactivation of the *gmhA* gene. Purine biosynthesis is known to be under the negative control of the PurR repressor protein [12]. According to our assumption, the deletion of the *purR* gene on the *ΔgmhA* background leads to suppression of sensitivity to antibiotics (Figure 1).

It is noteworthy that the deletion of the *purF* gene encoding amidoribosylphosphotransferase, an enzyme encoding the first stage of purine synthesis, abolishes the effect of *ΔpurR* (Figure 2), while deletion of *ΔpurH*, a phosphoribosylaminoimidazolecarboxamide formyltransferase/AICAR transformylase, one of the last enzymes of this pathway, has no such effect. These data indicate that the completion of most of the purine biosynthetic pathway up to the formation of AICAR (*purH* deletion) is needed to suppress the sensitivity of *gmhA* mutant to antibiotics.

### 2.2. Suppression of Oxidative Stress

We have previously demonstrated the development of oxidative stress in *E. coli* mutants with impaired synthesis of ADP-heptose [3]. One of the key protective mechanisms against oxidative stress is the activation of the *soxRS* system, which controls the expression of superoxide dismutase (*sodA*) genes, iron metabolism (*fur*), and the generation of reducing equivalents in the oxidative branch of PPP (*zwf*) [13]. Mutant strains carrying deletions of the genes *∆gmhA*, *∆purR*, and *∆gmhA**∆purR* were transformed by a plasmid carrying the *lux* operon of luminous bacteria, as reported under the control of the *soxS* promoter. Data of Figure 3a indicate a significant increase in the luminescence signal in the *gmhA* mutant, while the introduction of the *purR* deletion into the genome of this strain abolished this effect. The high level of oxidative stress characteristic of the *gmhA* mutant is thus effectively suppressed by the *purR* deletion.

Consistently with high activity of *soxS*-regulated promoters, a noticeable decrease in the level of reducing equivalents of NADPH is observed in the *gmhA* mutant. It should be stressed that the regulon controlled by the PurR repressor includes the serine–glycine pathway, in which the formation of 10-formyltetrahydrofolate, necessary for the synthesis of purines, is accompanied by the generation of NADPH equivalents. The restoration of the NADPH pool in the *∆gmhA∆purR* mutant (Figure 3b) seems to result from an increased metabolic flow in the serine–glycine pathway.

The presence of oxidative stress and a high level of generation of superoxide radicals in “deep rough” mutants leads to a high sensitivity to oxidants. Indeed, *ΔgmhA* strains exhibit a significant increase in sensitivity to paraquat (Figure 3c). Activation of purine biosynthesis, coupled with the generation of NADPH equivalents in the serine–glycine pathway, suppresses the sensitivity to paraquat of the *gmhA* mutant (Figure 3d).

The total ROS level in the double *∆gmhA∆purR* mutant remains as elevated as in the *∆gmhA* mutant and the percentage of dead cells in the population was even higher (Figure 4), whereas antibiotic sensitivity was reduced. Thus, in this case, antibiotic sensitivity does not seem to be related to overall ROS levels and percentage of dead cells. Therefore, the sensitivity to an antibiotic is not always correlated with an increase in overall ROS content and percentage of cell death in a population.

In this regard, it should be noted that, in cells that have lost ADP-heptose, the level of total ROS and the level of superoxide anion radicals can differ significantly [3]. The correlation with antibiotic sensitivity was observed for superoxide anion levels and not for the total ROS pool [3]. According to the data obtained (Figure 1, Figure 3 and Figure 4), the sensitivity of the studied strains to antibiotics correlates with the level of activity of the *soxS* promoter of the oxidative stress response system, and not with the total level of ROS. It is necessary to denote that the fluorescent probe for ROS (DHR123) does not react to the superoxide anion radical since superoxide is not an oxidizing agent but a reducing agent [14].

### 2.3. Oxidative Modification of Protein Thiol Groups

Evaluation of oxidative modification of protein thiol groups also demonstrated that cells with the *gmhA* deletion exhibit the highest level of oxidative damage to proteins, which is a consequence of oxidative stress (Figure 5). In the case of double deletion of *∆gmhA∆purR*, we observed a decrease in protein oxidation, indicating partial relief of oxidative stress in these cells.

Cells with *gmhA* deletion are characterized by a significant change in thiol redox status, in particular, an increase in the total intracellular pool of thiols [3] and a decrease in glutathione levels (Figure 6). Double deletion of *∆gmhA∆purR* only slightly reduces elevated thiol levels but results in a significant increase in glutathione level (Figure 6). The level of reduced glutathione (GSH) returns to the control value, while the level of oxidized glutathione (GSSG) exceeds the control value, which correlates well with the level of total ROS, which remains elevated (Figure 6b).

### 2.4. Glutathionylation of Proteins in the ΔgmhA Mutant

Disturbance of thiol homeostasis, including a decrease in the level of reduced glutathione and an increase in ROS level (Figure 5 and Figure 6) in the *gmhA* mutants may indicate the presence of such a redox modification of proteins as glutathionylation. Indeed, there is a significant increase in the level of protein glutathionylation for different proteins in *gmhA* mutant cells (Figure 7). The combined deletion of *purR* and *gmhA* results in drastic changes in the pattern of protein glutathionylation, and also leads to a significantly reduced level of glutathionylation of a number of proteins (Figure 7).

Assessment of the total level of protein glutathionylation (Figure 8a) also demonstrated that inactivation of *∆gmhA* in *E. coli* cells leads to an increase in the level of protein glutathionylation, while the combined deletion of *gmhA* with *∆purR* reduces overall glutathionylation as compared to control levels.

One of the examples of redox regulation of enzymatic activity by means of glutathionylation is aldolases of the first type involved in gluconeogenesis [15,16]. It is known that the activity of the first type of aldolase increases when glutathione binds to regulatory cysteine residues [16]. The determination of the total aldolase activity showed its significant increase in the cells of the *gmhA* mutant compared with wild-type bacteria, while *purR* deletion leads to normalization of the protein glutathionylation level and aldolase activity (Figure 8a,b). Accordingly, the inactivation of the *fbaB* gene encoding the first type of aldolase also leads to a decrease in aldolase activity in *gmhA* mutants, as compared with the wild-type strain (Figure 8b).

## 3. Discussion

Our data clearly demonstrate the effect of the intensity of the anabolic process of purine synthesis on antibiotic sensitivity and on the redox parameters of the *gmhA* mutant cells. It has been shown that the constitutive expression of the *purR*-regulon genes, including the serine–glycine pathway, leads to suppression of sensitivity to antibiotics (nalidixic acid and rifampicin) and oxidants (paraquat), as well as to normalization of the level of reducing equivalents of NADPH and the activity of the *soxRS* system and to a decrease in oxidative modification of the proteins (Figure 9a,b). It is controversial that the restoration of NADPH and glutathione levels is not accompanied by a normalization of ROS and the number of dead cells. However, it should be noted that purine biosynthesis requires large amounts of ATP and appears to provoke activation of the Krebs cycle and the respiratory chain, a source of ROS. On the other hand, the *soxRS* defense system, which regulates the activity of superoxide dismutase, senses the level of NADPH [17] and is inactivated when it is normalized. Thus, we observe an interesting phenomenon of suppression of sensitivity to antibiotics in the *ΔgmhAΔpurR* mutant compared to the *ΔgmhA* mutant, along with a decrease in viability.

The increased level of glutathionylated proteins found by us is an important indicator of the redox imbalance that occurs on the background of impaired synthesis of ADP-heptose. The change in enzyme activity by glutathionylation plays an important role in the regulation of biosynthetic processes [18]. An increase in the activity of FbaB aldolase in *gmhA* mutants on the background of an increased level of protein glutathionylation indicates a significant redistribution of central metabolic flows. In addition to aldolase, the enzyme of the non-oxidative branch of PPP—transketolase (Tkt) is also subject to such redox regulation. It is known that oxidative modification of Tkt cysteine residues can completely inactivate this enzyme [19]. We hypothesize that a change in the glutathionylation pattern of enzymes in *∆gmhA* mutants disrupts the processes of coupling the generation of NADPH equivalents with the synthesis of PPP intermediates, thereby reducing the stress resistance of cells. Thus, in summary, our data allow us to conclude that the derepression of purine biosynthesis normalizes the redox equilibrium in ADP-heptose biosynthesis mutants, reducing their sensitivity to antibacterial drugs.

## 4. Materials and Methods

**Bacterial strains**. The bacterial strains of *Escherichia coli* used in the work and their genotype are presented in the Table 1. Deletion mutants were obtained by growing phage P1 on strains from the Keio collection [20] containing the insertions *purR::kan, purF::kan, purH::kan, and fbaB::kan* and their subsequent transduction into the genome of the strain *E. coli* MG1655. The resulting strains were removed from the kanamycin cassette using helper plasmid pCP20 [21] with the formation of the corresponding deletions *∆purR*, *∆purF*, *∆purH*, and *∆fbaB*. The presence of deletions was confirmed by PCR experiments. Obtaining a strain containing a deletion of the *gmhA* gene is described in [3]. To quantify the level of activation of the *soxRS* genes of the system, a hybrid plasmid pSoxS’::lux was used, in which the promoter–operator region before the *soxS* gene was transcriptionally fused with the *luxCDABE P. Luminescens* gene cassette [22].

**Cultivation environments and conditions**. LB medium without glucose was used as a complete nutrient medium for growing bacteria [22]. In a liquid medium, the bacteria were cultured on a rocking chair (200 rpm). If necessary, nalidixic acid was added to the medium—0.5 µg/mL, rifampicin—1 µg/mL, ampicillin—50 µg/mL, kanamycin—20 µg/mL, paraquat—250 µM. All reagents used in the work produced by Sigma-Aldrich, St. Louis, MO, USA, unless otherwise stated.

**Generation of growth curves**. Growth curves were obtained on a Bioscreen C (Growth Curves Ltd., Turku, Finland) automated growth analysis system. Subcultures of specified strains were grown overnight at 37 °C, diluted 1:100 in fresh medium, inoculated into honeycomb wells in triplicate, and grown at 37 °C with maximum shaking on the platform of the Bioscreen C instrument. OD600 values were recorded automatically at specified times, and the means of the triplicate cultures were plotted.

**Determination of the sensitivity of bacteria to antibiotics and oxidants**. Overnight cultures of bacteria were diluted in fresh medium at 1:100 and grown at 37 °C with aeration to a titer of 10^7^, treated with the indicated concentrations of antibiotics and oxidants and continued to grow for 90 min. Then, dilutions were made and samples were sown on cups with LB medium, which were placed in a thermostat at 37 °C for 24 h. Survival was determined by counting colonies in three independent experiments to determine the average values. In addition, the ability to form colonies was evaluated by the microdilution method. Overnight bacterial cultures were diluted 100 times and grown on a thermostatic rocking chair at 37 °C to an optical density of OD_600_ = 0.5–0.6. All suspensions were aligned according to optical density. From the obtained cultures, a series of tenfold dilutions was prepared in a 96-well plate in the volume of 100 µL. The resulting dilutions were sown on cups with a rich medium containing various concentrations of the studied antibiotics. The cups were incubated overnight in a thermostat at 37 °C. The result was photographed using the GelCamera M-26XV Analytical system (UVP LLC, Upland, CA, USA, VisionWorks LS 7.0 Software).

**Measurement of the NADPH level**. The NADPH level was measured using a fluorimetric NADP/NADPH Assay Kit (Abcam, Cambridge, UK). The cells were grown up to OD_600_ ≈ 0.5 in a thermostated shaker at a temperature of 37 °C. The preparation of cell extracts, as well as all subsequent manipulations, were carried out according to the instructions of the manufacturer attached to the kit. The fluorescence of the samples was detected in a Tecan Spark tablet reader at Ex/Em = 540/590 nm. The results obtained were attributed to the optical density of the OD_600_ culture and expressed as a percentage. The level of NADPH in the wild type strain was taken as 100%.

**Measuring of the protein glutathionylation level**. Quantitative determination of the level of protein glutathionylation was carried out by the modified Titz method [23]. 100 µL of night culture was transferred to 10 mL of fresh LB medium and grown to OD_600_ ≈ 0.5. Cells from 5 mL of suspension were precipitated by centrifugation and suspended in a KPE lysing buffer with the addition of 0.1% Triton X100. The cells were homogenized with a pestle and centrifuged at 11,000 rpm for 5 min at 4 °C. The super was transferred to clean test tubes and treated with sodium borohydride as described in [23]. The obtained extracts were used to formulate the reaction. The reaction mixture, consisting of 20 µL of cell extract and equal volumes (60 µL) of DTNB, glutathione reductase, and NADPH, was incubated for 3 min at room temperature, after which adsorption was measured at 412 nm. GSH (Sigma Aldrich, St. Louis, MO, USA) was used to construct the calibration curve. The obtained values were attributed to the optical density of cultures. To determine the oxidized form of glutathione (GSSG), the obtained cell extracts were treated with 2-vinylpyridine as described in [23]. Oxidized glutathione was used to construct the calibration curve (Sigma Aldrich, St. Louis, MO, USA).

**Determination of the *soxS* promoter activity**. Overnight cultures of strains containing the pSox::lux plasmid were diluted to a concentration of 107 cells per 1 mL in fresh Luria–Bertani broth medium and grown under aeration at 30 °C until the early exponential growth phase; 200-μL aliquots were transferred to a 96-well tablet. The luminescence signal was detected on a Tecan Spark tablet reader for 3 h [22].

**Determination of Viability and Redox Status of Cells Using Flow Cytometry**. Viability and redox status of *E. coli* cells was determined using flow cytometry as we described earlier [3]. Cells were grown in a complete medium to an optical density of 0.4 and then washed twice with phosphate-buffered saline (1xPBS). They were then centrifuged, the supernatant was removed, and cells were resuspended in 100 µL of PBS. The cell population for analysis was selected according to the parameters of forward (FSC) and side scattering (SSC), which characterize the size and granularity of cells. The percentage of dead cells in the cell population was assessed using propidium iodide (Ex/Em = 535/617 nm, Sigma-Aldrich, St. Louis, MO, USA), which was added to the cells at a concentration of 10 µg/mL per minute before the start of the analysis. Propidium iodide penetrates the cells with a damaged membrane and, after binding to DNA, has a bright fluorescence in the red region of the spectrum. The redox status of E. coli cells was assessed by the level of reactive oxygen species (ROS) and intracellular thiols, a part of which is intracellular reduced glutathione. The ROS level was assessed using the dye Dihydrorhodamine 123 (DHR123) (Ex/Em = 507/525 nm, ThermoFisher Scientific, Waltham, MA, USA), which was added to the cells to a final concentration of 7.5 µM, and the cells were incubated for an hour in the dark at 37 °C. The levels of intracellular thiols were assessed using ThiolTracker Violet dye (Ex/Em = 405/526 nm, ThermoFisher Scientific), which was added at a concentration of 10 µM, after which the cells were incubated for one hour in the dark at 37 °C. These parameters were evaluated in the cells with intact membranes, which were not propidium iodide stained. Cell parameters were analyzed using flow cytometry on a BD LCR Fortessa flow cytometer (Becton Dickinson, Franklin Lakes, NJ, USA).

**Determination of protein glutathionylation and oxidation by immunoblotting**. The level of S-glutathionylation of *E. coli* proteins was estimated using immunoblotting. Cells from 10 mL of suspension were precipitated by centrifugation at 3000× *g* and washed twice with PBS. Cells were frozen in liquid nitrogen and then subjected to a threefold fast freezing (−196 °C) and thawing (37 °C) procedure. Then they were lysed in RIPA buffer (25 mM tris-HCl, pH 7.6, 150 mM NaCl, 1% Nonidet-P40, 0.1% SDS, 1% sodium deoxycholate) containing the protease inhibitors cocktail (Roche, Basel, Switzerland, 11836145001) and 5 µL PMSF (Roche, Basel, Switzerland, 6367.1) with stirring at 4 °C during one hour. The cell lysates were centrifuged at 16,000× *g* for 10 min at 4 °C and the supernatant was collected. Proteins of cell lysates were separated on 10% SDS-PAGE and transferred to a PVDF membrane. After the blocking procedure, mouse monoclonal anti-glutathione antibody (1:1000) (Chemicon Millipore, Burlington, MA, USA, MAB5310) or rabbit polyclonal antibody (1:1000) anti-cysteine sulfenic acid (Merck, Burlington, MA, USA, 07-2139-I) were added, followed by horseradish peroxidase-conjugated secondary antibodies. Membrane was stained using a commercial kit SuperSignal West Femto Maximum Sensitivity Substrate (ThermoFisher Scientific, Waltham, MA, USA) and chemiluminescence was detected using a Bio-Rad ChemiDoc MP Instrument (Bio-Rad, Hercules, CA, USA). Densitometric analysis was performed by the Image Lab (Bio-Rad, Hercules, CA, USA) program and the results were represented as ratio of glutathionylated protein or oxidized protein to total protein. The ratios of the bands (GSS-Pr)/total protein or (Pr-SOH)/total protein in the control were taken as 1. Protein was assessed using BCA Protein Assay Kits (ThermoFisher Scientific, Waltham, MA, USA).

**Aldolase activity assay**. Aldolase activity in cell extracts was determined using the commercially available Aldolase Activity Assay Kit (Colorimetric) (ab196994) (Abcam, Cambridge, UK). Cells were grown in a thermostatic shaker to OD_600_ ≈ 0.5 and collected by centrifugation. Then, they were washed with saline solution and the extract was prepared according to the factory instructions. The reaction was carried out according to the attached protocol. Data were obtained in three independent biological replicates.

### Statistical Analysis

The data are shown as mean ± standard deviation measures from triplicate values obtained from 3–4 independent experiments. The statistical difference between experimental groups was analyzed by one-way ANOVA with Tukey correction for multiple comparisons. Probability values (*p*) less than 0.05 were considered significant. Statistical analysis was performed using the GraphPad Prism 9.1.2 software (GraphPad Software Inc., San Diego, CA, USA).

## Figures and Tables

**Figure 1 ijms-24-16070-f001:**
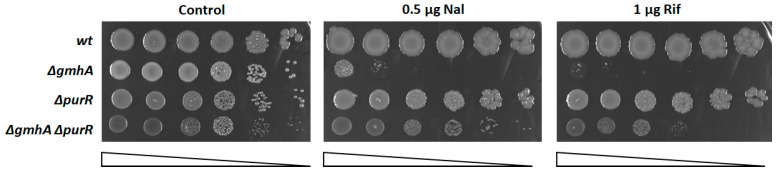
Deletion of the PurR repressor suppresses the sensitivity of *gmhA* mutants to the action of antibiotics. Representative efficiencies of colony formation of WT (MG1655) and mutant *E. coli* cells without (control) and in the presence of 0.5 µg nalidixic acid (Nal) or 1 µg rifampicin (Rif). Cells were spotted on LB agar plates in serial 10-fold dilutions (indicated by a triangle) and incubated at 37 °C for 24 h.

**Figure 2 ijms-24-16070-f002:**
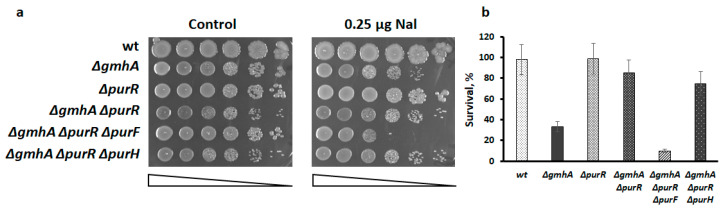
Interruption of purine synthesis at the stage of conversion of 5-phosphoribosyl-1-pyrophosphate to 5-phosphoribosyl-1-amine (*∆purF*) abolishes the suppressive effect of *purR* deletion. (**a**) Representative efficiencies of colony formation of WT (MG1655) and mutant *E. coli* cells without (control) or in the presence of 0.25 µg nalidixic acid (Nal). Cells were spotted on LB agar plates in serial 10-fold dilutions (indicated by a triangle) and incubated at 37 °C for 24 h. (**b**) Overnight cultures of indicated *E. coli* strains were diluted with fresh LB 1:100 and grown to ~2 × 10^7^. Nalidixic acid was added to reach 2.5 µg for 1 h. Cell survival was determined by counting cfu and is shown as the mean ± SD from three independent experiments. Deletion of *purF* reduces the survival of the *ΔgmhAΔpurR* strain in the presence of nalidixic acid, while *ΔpurH* has no such effect.

**Figure 3 ijms-24-16070-f003:**
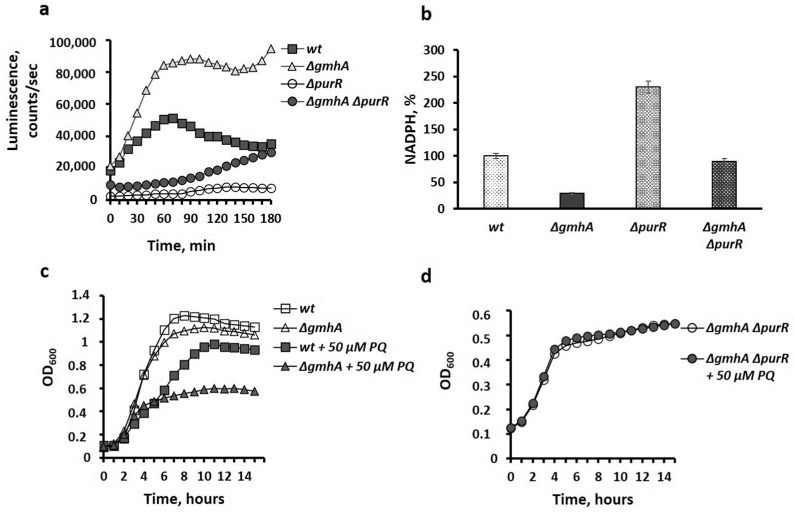
(**a**) Activity of the *soxS* promoter of the oxidative stress response system. (**b**) The level of reducing equivalents in the *gmhA* mutant on the background of *purR* deletion. Mean values ± SD from at least three independent experiments are shown. (**c**) Growth curves of *gmhA* mutant cultures in the presence of paraquat. (**d**) Suppression of sensitivity to paraquat in the strain *∆gmhA ∆purR*.

**Figure 4 ijms-24-16070-f004:**
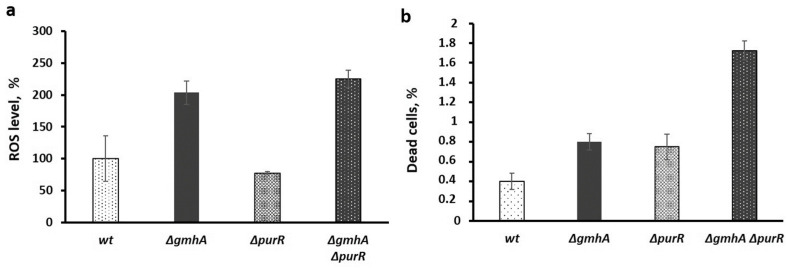
Changes in the level of intracellular ROS and the percentage of dead cells in the population of *E. coli* cells with deletions of the *gmhA* or *purR* genes or simultaneous deletion of *gmhA* and *purR*. (**a**) The level of ROS was assessed using the dye DHR123. (**b**) The percentage of dead cells was estimated using propidium iodide. Mean values ± SD from at least three independent experiments are shown.

**Figure 5 ijms-24-16070-f005:**
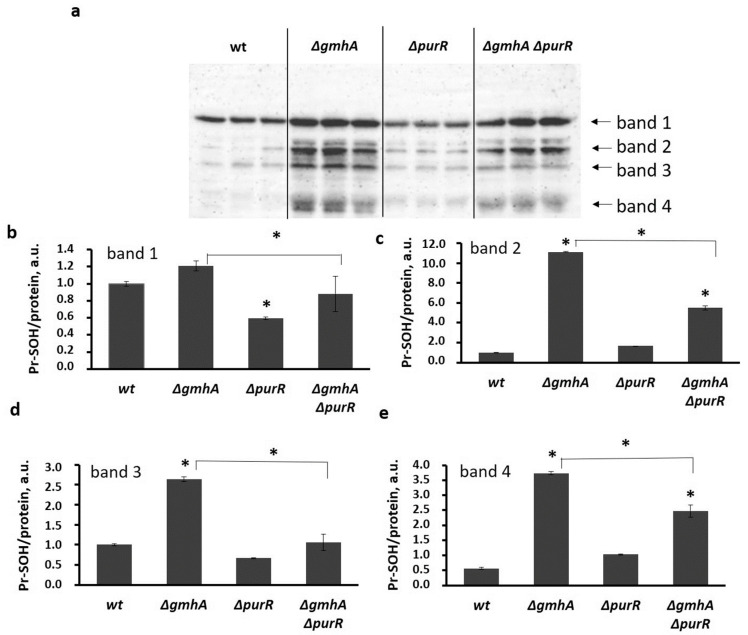
Oxidative modifications of protein thiol groups of *E. coli* proteins for cells of wide type and cells with deletions of the *gmhA* or *purR* genes and simultaneous deletion of *gmhA* and *purR*. (**a**) The original immunoblotting readouts. Oxidation of proteins was detected with anti-cysteine sulfenic acid. (**b**–**e**) Bars representing the oxidized (Pr-SOH) form of the proteins normalized by the total protein amount (Pr-SOH/protein) for band 1 (**b**), band 2 (**c**), band 3 (**d**), and band 4 (**e**), correspondingly. Mean values ± SD from three independent experiments are shown. * *p* < 0.05, compared to the wild-type cells and for *ΔgmhA* compared to the *ΔgmhA*
*ΔpurR*.

**Figure 6 ijms-24-16070-f006:**
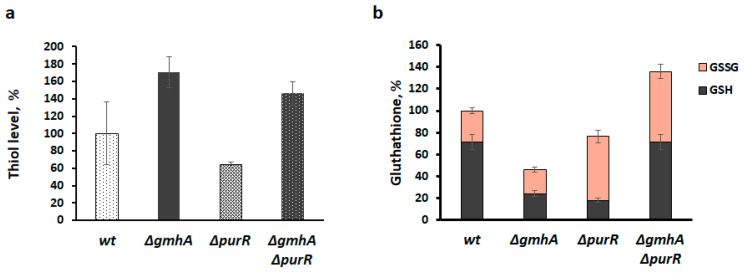
Changes in the level of intracellular thiols and the percentage of GSH/GSSG in the population of *E. coli* cells with deletions of the *gmhA* or *purR* genes and simultaneous deletion of *gmhA* and *purR*. (**a**) The total level of intracellular thiols was assessed using the dye ThiolTracker Violet. (**b**) The percentage of GSH/GSSG was estimated using the modified Titz method. Mean values ± SD from at least three independent experiments are shown.

**Figure 7 ijms-24-16070-f007:**
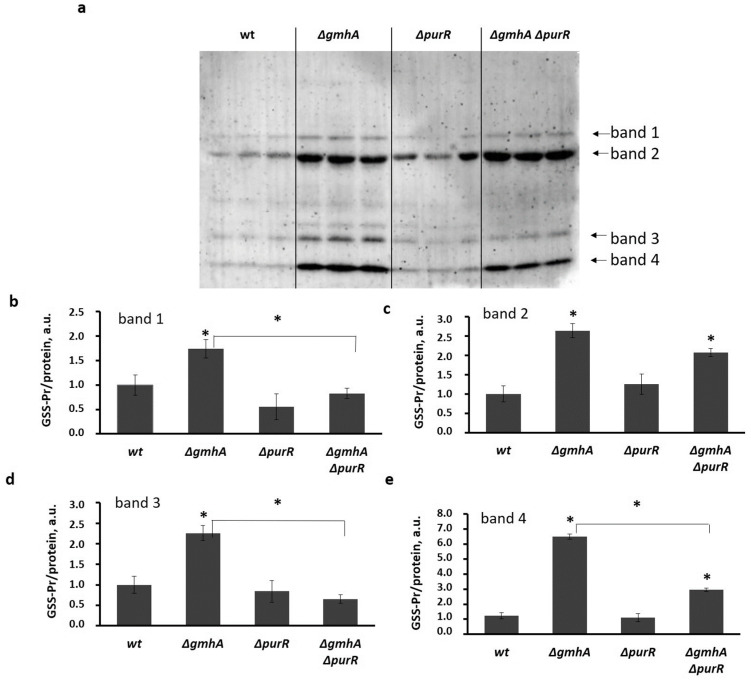
S-glutathionylation of *E. coli* proteins for cells of wide type and cells with deletions of the *gmhA* or *purR* genes and simultaneous deletion *gmhA* and *purR*. (**a**) The original immunoblotting readouts. Glutathionylation was detected with anti-glutathione antibodies. (**b**–**e**) Bars representing the S-glutathionylated (GSS-Pr) form of the proteins normalized by the total protein amount (GSS-Pr/protein) for band 1 (**b**), band 2 (**c**), band 3 (**d**), and band 4 (**e**), correspondingly. Mean values ± SD from three independent experiments are shown. * *p* < 0.05, compared to the wild-type cells and for *ΔgmhA* compared to the *ΔgmhA*
*ΔpurR*.

**Figure 8 ijms-24-16070-f008:**
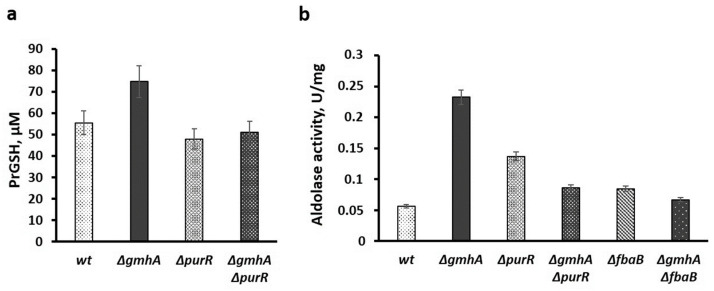
The level of protein glutathionation (**a**) and aldolase activity (**b**) in *gmhA* mutants against the background of *purR* and *fbaB* gene deletion. Mean values ± SD from three independent experiments are shown.

**Figure 9 ijms-24-16070-f009:**
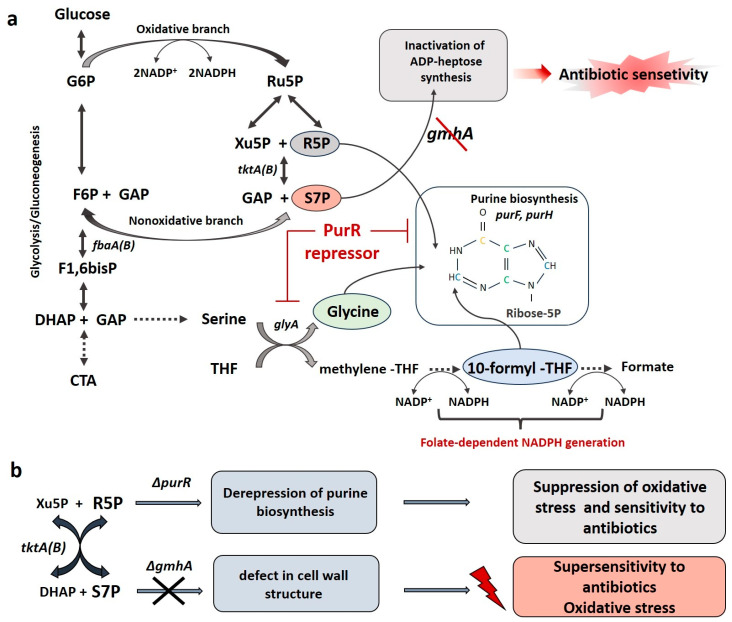
(**a**) Metabolic scheme of the PPP, demonstrating the relationship between the processes of synthesis of the ADP-heptose precursor—sedoheptulose-7-phosphate, ribose-5-phosphate, and the reduction of NADP^+^. The PurR repressor regulates the synthesis of purines, starting with the formation of phosphoribosyl pyrophosphate, as well as the serine–glycine pathway (*glyA* gene), which, along with the oxidative branch of PPP, is a source of NADPH generation. (**b**) Inactivation of ADP-heptose synthesis leads to the “deep rough” phenotype, hypersensitivity to antibiotics and oxidative stress. Activation of purine biosynthesis in bacterial cells with impaired synthesis of ADP-heptose suppresses antibiotic sensitivity and oxidative stress.

**Table 1 ijms-24-16070-t001:** Genotype and origin of *E. coli* strains.

Strain	Genotype	Origin
MG1655	F– wild type	[4]
*∆gmhA*	Deletion of the *gmhA* D-sedoheptulose 7-phosphate isomerase	[3]
*∆purR*	Deletion of the DNA-binding transcriptional repressor PurR	This work
*∆gmhA ∆purR*	Deletion of the *gmhA* D-sedoheptulose 7-phosphate isomerase and deletion of the DNA-binding transcriptional repressor PurR	This work
*∆gmhA ∆purR ∆purF*	Deletion of the *gmhA* D-sedoheptulose 7-phosphate isomerase, deletion of the DNA-binding transcriptional repressor PurR and deletion of the amidophosphoribosyltransferase	This work
*∆gmhA ∆purR ∆purH*	Deletion of the *gmhA* D-sedoheptulose 7-phosphate isomerase, deletion of the DNA-binding transcriptional repressor PurR and deletion of the bifunctional AICAR transformylase/IMP cyclohydrolase	This work
*∆fbaB*	Deletion of the fructose-bisphosphate aldolase class I	This work
*∆gmhA ∆fbaB*	Deletion of the *gmhA* D-sedoheptulose 7-phosphate isomerase and deletion of the fructose-bisphosphate aldolase class I	This work
MG1655 pSoxS::lux	As MG1655, plus pSox::lux plasmid	This work
*∆gmhA* pSoxS::lux	As *∆gmhA*, plus pSox::lux plasmid	This work
*∆purR* pSoxS::lux	As *∆purR*, plus pSox::lux plasmid	This work
*∆gmhA ∆purR* pSoxS::lux	As *∆gmhA ∆purR*, plus pSox::lux plasmid	This work

## Data Availability

This study did not report any data.

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
