# Peer review of "Activation of Purine Biosynthesis Suppresses the Sensitivity of E. coli gmhA Mutant to Antibiotics"

_ijms, 2023, doi:10.3390/ijms242216070_

Round 1
Reviewer 1 Report
Comments and Suggestions for Authors
here my comments:
In the introduction it would be useful to have a schematic representation of the metabolic pathway being discussed, this would facilitate reading.
figure 1 represents the effect of two compounds on cells, there is something I am not clear about in the representation though: the experiment is conducted at 0.5ug of NaI, what is NaI? and as a positive control, I believe there is 1ug of Ref. What is that? What happens in the control? Also, below the figures, there is a triangle representing a decrease of something. What exactly? This figure is not clear at all.
figure 2a same problem as the previous figure
please do not use the word 'speculate' replace it with suppose or hypothesize
Author Response
We thank you for your careful reading of our work and valuable comments.

Reviewer 2 Report
Comments and Suggestions for Authors
The manuscript of Tatiana A. SereginaI et al. entitled “ Activation of purine biosynthesis suppresses the sensitivity of E. coli gmhA mutant to antibiotics” reveals the balancing of the negative effects of the gmhA mutation, that are the sensitivity to antibiotics and oxidants, by over-expression of the purine biosynthetic pathways linked to the deletion of the gene encoding PurR, the repressor of this pathway. The deletion of gmhA leads to a reduction of NADPH and glutathion availability whereas the over-expression of the purine biosynthetic pathways restores the availability of these two molecules that are known to be involved in the resistance to oxidative stress. Indeed NADPH is a well known necessary co-factor of thioredoxins, enzymes, playing a crucial role in the resistance to oxidative stress. Glutathione (GSH) is a cellular tripeptide (L-γ-glutamate-L-cysteinyl-glycine) whose enhanced biosynthesis in the PurR mutant might be linked to the over-expression of the glycine –serine pathway in this mutant. Indeed since glycine enters in the constitution of glutathion, enhanced glycine biosynthesis is likely to have a positive impact on glutathione biosynthesis.
Overall the manuscript of Tatiana A. SereginaI et al. is interesting, the data are convincing and the hypothesis reasonable.
However the abstract does do summarizes accurately the content of the manuscript. The over-expression of the serine-glycine pathways in the PurR mutant should be mentioned since enhanced NADPH and glutathione biosynthesis is likely to be likely to this over-expression.
Even of the paper is rather clearly written some editorial modifications are needed. The reviewer proposed some modifications (in red) that should be carefully checked by the authors and some clarifications are also required (yellow parts)

Even if the paper is rather clearly written some editorial modifications are needed. The reviewer proposed some modifications (in red) that should be carefully checked by the authors and some clarifications are also required (yellow parts)
Author Response

(The authors gave the same response as above.)

Round 2
Reviewer 1 Report
Comments and Suggestions for Authors
No others comments from my side